# Light-Activable Silver Nanoparticles for Combatting Antibiotic-Resistant Bacteria and Biofilms

**DOI:** 10.3390/molecules30030626

**Published:** 2025-01-31

**Authors:** Varsha Godakhindi, Elana Kravitz, Juan Luis Vivero-Escoto

**Affiliations:** 1Department of Chemistry, University of North Carolina at Charlotte, Charlotte, NC 28223, USA; vgodakhi@charlotte.edu (V.G.); elana.kravitz@mail.huji.ac.il (E.K.); 2Nanoscale Science Program, University of North Carolina at Charlotte, Charlotte, NC 28223, USA; 3Center for Innovation, Translational Research and Applications of Nanostructured Systems, Charlotte, NC 28223, USA

**Keywords:** silver nanoparticles, photosensitizers, photodynamic inactivation, light activated antibacterials, antibiofilm agents

## Abstract

Silver nanoparticles (AgNPs) are among the most widely used nanoparticulate materials for antimicrobial applications. The innate antibacterial properties of AgNPs are closely associated with the release of silver ions (Ag^+^) and the generation of reactive oxygen species (ROS). Multiple reports have elaborated on the synergistic effect against bacteria by combining photosensitizers with AgNPs (PS-AgNPs). This combination allows for the light-activated generation of Ag^+^ and ROS from PS-AgNPs. This is an efficient and controlled approach for the effective elimination of pathogens and associated biofilms. This review summarizes the design and synthetic strategies to produce PS-AgNPs reported in the literature. First, we explore multiple bacterial cell death mechanisms associated with AgNPs and possible pathways for resistance against AgNPs and Ag^+^. The next sections summarize the recent findings on the design and application of PS-AgNPs for the inactivation of resistant and non-resistant bacterial strains as well as the elimination and inhibition of biofilms. Finally, the review describes major outcomes in the field and provides a perspective on the future applications of this burgeoning area of research.

## 1. Introduction

The declining efficacy of antibiotics and the resulting rise in antibiotic-resistant bacteria (ARB) have become severe public health problems. The overuse and misuse of antibiotics have resulted in multidrug-resistant (MDR) “superbugs”. Infections caused by MDR bacteria have remarkably increased the morbidity and mortality of patients, which are becoming huge challenges in clinical treatment. Therefore, it is urgently required to develop effective strategies with high antibacterial efficiency and a low drug-resistance risk for bacterial elimination. Several mechanisms have been associated with the resistance of ARB, including (a) the active efflux of antibiotics via efflux pumps, (b) a decrease in bacterial wall permeability, (c) modification of the antibiotic pathway or targets, and (d) the overexpression of antibiotic-resistant genes (ARGs) and subsequent enzymes [1,2,3].

Nanoparticles (NPs) have recently aroused great interest as a treatment option for ARB, owing to their small size, large surface-area-to-volume ratio, tunable surface properties, and unique optical properties [4,5,6]. The NP sizes are similar to those of biomolecules, allowing a range of interactions with bacteria that can be controlled through surface functionalization [7]. These NPs can afford a broad spectrum of killing mechanisms against bacteria with a lower risk of inducing bacterial resistance [5,6,7,8]. The antibacterial mechanisms of NPs can be defined as two general categories: (a) NPs with innate antibacterial properties, and (b) NPs used as carriers for delivering one or more antimicrobial agents [7,9]. The innate antibacterial properties of NPs rely on close interaction between them and bacteria. These bacterial death mechanisms include oxidative stress, metal-ion release, and non-oxidative mechanisms such as photothermal, ultrasonic, and magnetic effects [5,6]. The most studied materials are metal and metal oxide nanoparticles such as silver, gold, zinc oxide (ZnO), and copper oxide (CuO). Nevertheless, silver nanoparticles (AgNPs) have attracted more interest due to their small size (<100 nm), optical properties, and innate antibacterial properties [10].

The innate antibacterial properties of silver-containing molecules have been known since ancient times [11]. In particular, silver salts that generate silver cations (Ag^+^) are highly reactive species that afford several mechanisms to eliminate bacteria. For example, Ag^+^ can effectively bind to various internal proteins, inducing structural changes within the bacterial and cell membranes. Moreover, Ag^+^ can also interfere with the DNA replication process. Finally, Ag^+^ also induces oxidative stress [11,12]. AgNPs offer unique advantages over silver compounds, such as an increased concentration of Ag in situ, unique optical and catalytic properties, the controlled release of Ag^+^, and a targeting ability rendered by the surface functionalization of the NPs [11,12,13,14]. Therefore, it is envisioned that AgNPs have a higher antibacterial activity than silver molecules [13,15]. AgNPs kill bacteria via multiple mechanisms, such as disrupting the bacterial cell wall, interfering with the metabolic processes, deactivating the enzymes, causing DNA damage, and generating ROS, thereby increasing oxidative stress [11,14]. Some of these bacterial death mechanisms are similar to those related to Ag^+^; in fact, the bactericidal properties of AgNPs are highly associated with the efficient release of Ag^+^ ions [10,14]. Additionally, possible side effects related to the release of high concentrations of Ag^+^ ions must be reduced by a reliable, controlled delivery approach [15,16].

One approach recently explored how to control the release of Ag^+^ from AgNPs by combining the NPs with photosensitizers (PS). PSs are molecules that can be excited with light at a specific wavelength for its absorbance peak, the energy of which is transferred to oxygen to generate singlet oxygen and reactive oxygen species (ROS); this process is called photodynamic inactivation (PDI) [17,18,19]. Therefore, the combination of AgNPs with PSs offers a remote-control approach for releasing Ag^+^ through the oxidation of the AgNP surface mediated by ROS generation [20,21]. This alternatively allows for the light-activated generation of Ag^+^ from AgNPs and is an effective way to control the amount of Ag^+^ released and for the multiple releases of Ag^+^ at will. This combination is also known to elevate ROS generation. ROS generation induces an antibacterial effect, attacking the cellular membrane, essential enzymes, and cellular nucleic acids, leading to bacterial cell death by oxidative stress [22,23]. Finally, it is envisioned that by combining AgNPs with PDI, there is little probability of the bacteria developing resistance against this approach [5].

This review summarizes the PS and AgNPs design and the synthetic strategies reported in the literature for the inactivation of resistant and non-resistant bacterial strains and the elimination/inhibition of biofilms. A description of the intrinsic antimicrobial properties of AgNPs and their synergistic interaction with light to eliminate pathogens is provided. Different strategies for combining PS with AgNPs to eliminate bacteria and biofilms are summarized. Finally, conclusions and perspectives on the future directions for this field are illustrated. We expect that this review will give the audience, who are interested in using AgNPs to eliminate ARB and biofilms, a new vision of the potential possibilities of this nanomaterial.

## 2. AgNPs and Their Antimicrobial Properties

The synthesis of AgNPs can be achieved by reducing silver salts to generate stable AgNPs of variable nanostructures [14,24]. This approach requires a silver precursor, a reducing agent (chemical or biological), and a stabilizing agent to generate colloidal AgNPs. The chemical-reducing agents include sodium citrate, sodium borohydride, ascorbic acid, tannic acid, and glucose [25]. Biological reducing agents such as bacteria, fungi, and plant extracts provide an avenue to explore environmentally friendly synthetic methods [12].

Several studies have shown that the antibacterial properties of AgNPs are multifactorial, involving several mechanisms (Figure 1), including (a) the ability of AgNPs to anchor and penetrate the cell wall and membrane of bacteria, (b) the generation of intracellular ROS, (c) damaging the membrane-bound and intracellular sulfur- and phosphorus-containing groups such as enzymes and DNA, (d) the alteration of efflux pumps, (e) interfering with bacterial electron transport chain and (f) intercalating with DNA and RNA subunits [11,14,26]. As part of these bacterial death mechanisms, the bactericidal properties of AgNPs are also highly associated with the release of silver ions (Ag^+^) [11,27]. This has made some scientists hypothesize that bacteria can develop resistance against AgNPs. It has been shown that specific pathogens can eliminate Ag^+^ ions via efflux pumps or through the reduction of Ag^+^ ions to its less toxic form of elemental Ag [28,29]. However, as described above in Figure 1, AgNPs use mechanisms other than releasing Ag^+^ to eliminate bacteria. Recently, it was observed in Gram-negative bacteria that repeated exposure to AgNPs led to the development of adhesive flagella that induced aggregation of the nanoparticles, reducing their antibacterial effect [30,31]. Nevertheless, unlike antibiotics, the AgNP resistance mechanism in bacteria has not yet been associated with a genetic origin.

The physicochemical properties of AgNPs, such as size, shape, and surface charge, play a crucial role in their antibacterial action. Several studies have confirmed that smaller AgNPs of size < 30 nm have a superior antibacterial effect than larger AgNPs [11,12,14,32]. Some reports have correlated the size (5–30 nm) to the dissolution of AgNPs into Ag^+^ ions, which enhances the antibacterial action of AgNPs [10,14,33]. Other studies have suggested that AgNPs with cubic, plate, and rod shapes perform better as antibacterial agents than their spherical counterparts due to the presence of high atom density facets [10,33,34,35]. Nevertheless, other reports have suggested that AgNPs with spherical shapes and increased surface area result in a higher release of Ag^+^, making these nanoparticles better antibacterial candidates [36,37]. Therefore, the impact of the AgNPs’ shape on their bactericidal activity is still under investigation. On the other hand, AgNPs with positive surfaces have shown a more significant antibacterial effect, presumably due to the increased interaction of AgNPs with the surface of the bacteria [14]. It should be noted that the colloidal stability of AgNPs also plays a crucial role wherein lower stability is associated with aggregation, which subsequently reduces the antibacterial activity of AgNPs [38,39].

AgNPs are among the most used materials worldwide due to their well-known antibacterial activity, which makes them a promising alternative to antibiotics. However, there is a gap in the literature on the exact antibacterial mechanisms of AgNPs. Moreover, bacterial resistance against AgNPs is growing and is a significant concern. Therefore, there is a need to develop novel technologies that combine AgNPs with other approaches to eliminate pathogens efficiently.

## 3. Light-Mediated Killing of Pathogens Using AgNPs

AgNPs are plasmonic nanoparticles that can generate free electrons post-visible light irradiation as a result of their surface plasmon resonance (SPR) effect [40,41,42]. The SPR wavelength depends on the nanoparticle’s size and shape; in the case of AgNPs, this is in the 400–500 nm range. Researchers have generated different shapes of AgNPs to extend this wavelength range to near-infrared (NIR) regions. Bourgonje et al., in a recent report, demonstrated the phototherapeutic activity of triangular AgNPs (tAgNPs) and decahedral AgNPs (dAgNPs) under NIR and blue LED light [40]. These LED lights were selected to overlap with the SPR peak of the respective AgNPs, i.e., tAgNPs showed high absorbance at 800 nm and dAgNPs showed peak absorbance at 500 nm. In *S. aureus* and *E. coli*, tAgNPs showed an exceptional >9 log reduction in bacteria at 20 ppm post-1 min and 15 min of NIR irradiation, respectively. This antibacterial action was dependent on the concentration and irradiation time, with ~4 log and ~9 log reduction achieved for 5 ppm and 10 ppm after 15 min of NIR irradiation. In *E. coli*, dAgNPs reached only a 5 log reduction at 20 ppm concentration after 1 min of blue light irradiation, whereas dAgNP and tAgNP performed the same in *S. aureus* under the same conditions. Interestingly, tAgNPs and dAgNPs were unresponsive at light sources complementary to their SPR absorption wavelengths. The authors also monitored Ag^+^ release post-LED irradiation, wherein tAgNPs showed slightly higher leaching than dAgNPs. However, the levels of Ag^+^ (0.21 ppm) were too low to contribute to antibacterial action. The fluorescent probes for reactive oxygen and nitrogen species (ROS/RNS) indicated increased emission under NIR irradiation. Additionally, the antibacterial response tested in anaerobic conditions showed a reduced response (~2 log reduction). The decrease in antibacterial action in anaerobic conditions points to the importance of oxygen driving the antibacterial mechanism of these NPs. Thus, the authors concluded that the potential antimicrobial mechanism of NIR-activated tAgNPs was associated with increased ROS/RNS generation and plasmonic heating. Vasilkov et al. reported that the laser excitation of AgNP composites (bandages) at the SPR wavelength led to 15–24% CFU reduction in bacterial species [43]. This could be attributed to SPR-associated electronic changes in AgNP or localized heating that enhanced their antibacterial effect. Like most noble metals, the SPR in AgNP induces elevated absorption of the incident light, which is associated with ROS generation [44].

## 4. Combination of Photosensitizers with Silver Nanoparticles for Antibacterial Applications

Photosensitizers (PSs) are typically organic molecules that, after irradiation with light at the correct wavelength in the presence of molecular oxygen, generate reactive oxygen species (ROS) such as singlet oxygen (^1^O_2_), superoxide (O^2−^), and hydroxyl radicals (OH) [19,45,46,47]. The ROS and ^1^O_2_ are highly reactive species that can attack the cellular membrane, essential enzymes, and cellular nucleic acids [22,23]. The broad-spectrum antibacterial action of PSs serves as an advantage over antibiotics since the development of resistance to photochemical reactions is unlikely [19]. The use of PSs to eliminate pathogens is called photodynamic inactivation (PDI).

Silver nanoparticles have been physically or chemically combined with PSs such as porphyrins, phthalocyanines, and methylene blue [48,49,50,51]. These PS-AgNP combinations serve three main purposes: (a) increase the bioavailability of water-insoluble PSs, (b) tune the optical properties for PDI via plasmonic coupling with AgNPs or aggregation-induced emission (AIE), and (c) control the oxidation of the AgNP surface to enhance the release of Ag^+^ ions. In the first role, AgNPs are used as the delivery vehicles of PSs, increasing their bioavailability for effective PDI performance at low concentrations [52]. In the second function, AgNPs can tune the optical absorption of PSs according to the electronic overlap between the SPR of the AgNPs and the absorption band of the PS [53,54,55,56]. The third role deals with the oxidation effect of singlet oxygen and/or ROS on the surface of AgNPs to afford Ag^+^ ions, which improves the antibacterial outcome of the system [20,21,57,58,59]. Overall, it has been demonstrated that a synergistic effect exists in the combination of PS-AgNPs that improves their antibacterial effect [40,41,54].

The combination of PSs and AgNPs in the same system, reported in the literature, can be divided into four main categories (Figure 2): (1) PSs physically associated with AgNPs, (2) PSs chemically conjugated to AgNPs, (3) PSs and AgNPs incorporated into a nano platform (PS-AgNP nanocomposites), and (4) the sequential approach of PSs and AgNPs. A summary of all the different combinations of PSs and AgNPs reported in the literature is given in Table 1. In the following sections, some representative examples of each category are described.

### 4.1. PSs Are Physically Associated with AgNPs

The association of AgNPs and PSs is carried out through different molecular interactions between the surface of the nanoparticles and PSs, including electrostatic and hydrophobic interactions, which result in the adsorption of PS molecules onto the AgNPs’ surface. There have been multiple reports that show PS molecules physically linked to AgNPs show antibacterial synergy [63,66,68]. Li et al. reported a study elaborating on the synergy of AgNP/ Ag^+^ ions and methylene blue (MB) in five bacterial strains: *Serratia marcescens* (SM), *Escherichia coli*, *Klebsiella pneumoniae* (KP), Pseudomonas *aeruginosa* (PA), and *Enterobacter cloacae* (EC) as proof of the relevance of this combination [63]. The authors argued that the mechanism behind this synergy was an Ag^+^ interaction with the bacterial cell membrane that facilitated better MB uptake.

Elashnikov, Lyutakov, and co-workers have reported two studies wherein meso-tetraphenylporphyrin (TPP) was combined with silver nitrate followed by further interaction with polymethylmethacrylate (PMMA) nanofibers via electrospinning [66,68]. The presence of AgNPs was confirmed via TEM images. The authors first tracked the changes in the absorbance of the soret band of TPP under illumination (405 nm, 110 mW) and concluded that AgNPs/TPP/PMMA demonstrated higher photostability than only TPP/PMMA. These results implied that the AgNPs protected the TPP, within the PMMA nanofibers, against photobleaching. The antibacterial tests performed on *E. faecalis* and *S. epidermidis* showed that the AgNPs/TPP/PMMA nanofibers exhibited a significantly higher reduction in bacteria than TPP/PMMA. The disk-diffusion method showed <100 mm^2^ for TPP/PMMA compared to >400 mm^2^ for AgNPs/TPP/PMMA. This result was associated with the cumulative release of AgNP and the production of ROS. In a similar study, the authors argued that this release of AgNPs from the nanofibers was promoted by the illumination of TPP, inducing heat-related changes in the polymer and contributing to the escape of AgNPs.

Peng and co-workers have developed a hybrid core-shell nanoparticle with silver as the core and mesoporous silica as the shell (Ag@MS). Several photosensitizers such as mesoporphyrin IX (PIX), hematoporphyrin IX (HPIX), meso-tetra(4-carboxyphenyl) porphine (TCPP), Cu(II) mesotetra(4-carboxyphenyl) porphine (Cu-TCPP), tris(2,20-bipyridyl) dichlororuthenium(II) hexahydrate (RuBPy), and Rose Bengal (RB) have been adsorbed to the surface of Ag@MS [54,60,61]. Among these PS, all except RB show stronger resonance coupling with the AgNP core that was positively correlated with the enhanced production of singlet oxygen [54,61].

Lismont et al. validated this claim wherein PpIX loaded on Ag@MS NP with a larger AgNP core (100 nm) and a thin silica coating (5 nm) led to a 5-fold increase in singlet oxygen generation than PpIX [50]. The Ag@MS@HPIX hybrid showed increased singlet oxygen generation and increased antibacterial killing compared to its PS counterpart. In *MRSA*, these Ag@MS@HPIX hybrid nanoparticles resulted in up to ~6 log inactivation at a concentration equivalent to 2 µM of HPIX. In *S. epidermidis*, 5 log bacterial killing was seen at a HPIX concentration of 2 µM. While in both *E. coli* and *A. baumannii*, 4 log inactivation was observed at a HPIX concentration of 1 µM [54].

### 4.2. PS Chemically Conjugated to AgNPs

Unlike the approach of physical interaction between PSs and AgNPs, the chemical conjugation between PSs and AgNPs ensures stable nano-formulations with minimal leaking. A chemical bond between PS and AgNP also allows the control of distance and target delivery to enhance antibacterial synergy [50]. Most reports in the literature have utilized cysteinyl groups, NHS-EDC chemistry, thiol-Ag affinity, polymer-assisted self-assembly, and coordination via COOH and NH_2_ groups. Below, we provide representative examples of how the chemical conjugation approach has been used to treat pathogens.

Masilela et al. reported on the antimicrobial response of PS-conjugated spherical, triangular, and cubic AgNPs against *S. aureus*. Phthalocyanines were selected as the PS to utilize the amines (NH_2_) and carboxylic group (COOH) of the cysteinyl moiety for coordination with the AgNP surface. All of the PS-AgNP conjugates generated an equivalent amount of singlet oxygen; however, the spherically shaped AgNPs gave the highest antimicrobial activity compared to the triangular and cubic structures [48]. The authors accounted for this result based on the increased surface area provided by the spherical AgNPs due to their smaller size (15 nm) compared to the triangular (54 nm) and cubic (60 nm) AgNPs. This could be due to increased Ag^+^ release associated with the higher surface area of spherical AgNPs, which contributed to their antibacterial action [36].

Elashnikov and co-workers also synthesized chemically conjugated PS-AgNP systems [49]. These researchers generated porphyrin-AgNPs by reacting thiolated porphyrin on the surface of AgNPs. To achieve this, the authors synthesized 2,7,12,18-tetramethyl-3,8-divinyl-13,17-[bis((4-aminothiophenyl)-carbamoyl]ethyl) porphyrin via an amide coupling procedure between the free carboxylic acid groups of protoporphyrin IX (PpIX) and the aromatic amine with thiol groups from 4-aminothiophenol. Under 1 h of blue LED exposure, this nanoplatform was successfully able to eradicate *S. epidermidis* and *E. coli* completely. This study argued that the ROS generation was enhanced due to the inclusion of AgNPs, leading to an elevated antibacterial effect. This could be associated with the plasmonic coupling of AgNP with PpIX, causing an increase in ROS [53,54,55,56].

Another crucial aspect of the light-activated PS-AgNPs is the release of Ag^+^ cations associated with the oxidation of the AgNPs surface in the presence of PS-generated ROS. However, only a couple of studies have elaborated on the porphyrin-mediated enhanced release of Ag^+^ ions and their corresponding antibacterial effect [21,59]. Sorinolu and co-workers reported an in-depth study detailing the release kinetics of Ag^+^ post-light irradiation to understand their role in the antibacterial inactivation of PpIX-AgNP conjugates [21]. Cysteamine-modified PpIX derivatives were synthesized, followed by the thiol-Ag reaction to generate PpIX-AgNP conjugates. The results indicated that PpIX-AgNPs showed higher Ag^+^ release than AgNPs. Due to its ionic composition, this effect was more pronounced (25-fold) in Dulbecco’s phosphate buffer solution (DPBS) than in Nanopure water. The authors hypothesized that the presence of Cl^-^ ions could account for some of the Ag^+^ release differences between water and DPBS. The study further proved that this light-mediated increase in Ag^+^ contributed to the antibacterial synergy resulting in up to the ~7–8 log inactivation of methicillin-resistant *Staphylococcus aureus* (MRSA) and wild-type multidrug-resistant (MDR) *E. coli* in DPBS.

### 4.3. PS and AgNP Incorporated into Nanoplatforms (PS-AgNP Nanocomposites)

Another approach to developing light-activated AgNP-based systems is incorporating PS-AgNP conjugates, physically or chemically loaded with PSs, into hybrid platforms at the nano or microscale. These systems have been termed PS-AgNP nanocomposites. These nanocomposites include fabrics, doped nanomaterials, and polymeric nanomaterials [58,59,67]. This alternative confers unique benefits such as stability, target delivery, and potential multi-modal antibacterial applications.

Chen et al. developed an antimicrobial cellulose-based fabric embedded with AgNPs and zinc-phthalocyanines (PS) [59]. The PS was first mixed with DMF solution containing N, N’-carbonyldiimidazole, followed by the introduction of cellulose fabric into the solution to obtain PS/Fabric via an ester bond. This PS/Fabric was further mixed with Ag^+^ ions which are reduced in the presence of the hydroxyl groups contained in the cellulose fabric to generate AgNPs (AgNP/PS/fabric). SEM studies showed that the sizes of the AgNPs embedded in the fabric are ~100 nm in diameter. Moreover, the UV-Vis spectrum depicted peaks at 420 nm and 600–800 nm, which are associated with AgNPs and the Q-band of phthalocyanines, respectively. The authors tested the physicochemical properties of the system wherein AgNP/PS/fabric demonstrated a 2-fold higher Ag^+^ ion release under light conditions than dark conditions. In addition, a higher increase in the ROS generation by the AgNP/PS/fabric system than PS/fabric was also observed. These results corroborated that the AgNP/PS/fabric was capable of producing both Ag^+^ and ROS. The antimicrobial properties of the material were evaluated: PS/fabric showed only 7.56% inhibition whereas the AgNP/PS/fabric resulted in a significant 3 log or 99.996% inhibition in *E. coli*, *S. aureus* (non-resistant), and *MRSA* under 10 min irradiation with light. This further provided evidence to support the broad-spectrum application of PS-AgNPs.

Kuthati et al. employed a similar nanocomposite design to demonstrate the synergy between silver nanoparticles (AgNP) and PS by combining AgNP-loaded Cu-MSN with curcumin [67]. Three antimicrobial agents played a crucial role in the synergy in this design, i.e., copper ions, silver nanoparticles, and curcumin. The design improved antibacterial activity since curcumin delivery was enhanced using MSN carriers and the positive charge of the AgNP-loaded Cu-MSN facilitated better interaction with the bacteria. The Cur-Cu-MSN-AgNP hybrid with a curcumin concentration of 1.5 µM resulted in a ~90% (5 log higher than curcumin only) eradication of Gram-negative *E. coli*. The authors concluded that blue LED light-mediated bactericidal synergy against *E. coli* was effective due to elevated Ag^+^ ion release and ROS generation.

Xu and collaborators developed a photoactive nanocomposite containing AgNPs, chlorin e6 (Ce6), and bacteria-targeting ligands to allow the light-controlled elimination of *MRSA* and *E. coli* [58]. The surface of the AgNPs was modified with a polydopamine shell (PDA) to load the Ce6 molecules to obtain Ce6-AgNPs. Following this, a bacteria-targeting ligand (GP) was tethered to the PDA shell of the Ce6-AgNPs via a PEG linker containing amine and thiol groups to obtain the final nanocomposites i.e., GP-Ce6-AgNPs (~50 nm size, ~4 nm PDA shell). The authors demonstrated that the continuous 30 min laser irradiation of the GP-Ce6-AgNPs resulted in a gradual release of Ag^+^ ions. When these nanocomposites were exposed to short laser pulses in an ON (5 min) and OFF (5–10 min) pattern, the release of Ag^+^ ions increased only during ON, whereas the amount of Ag^+^ reached saturation during OFF, followed by a similar pattern in the other two irradiation cycles. The in vitro relative antibacterial rate for GP-Ce6-AgNP in *MRSA* and *E. coli* was reported as 99.6% and 98.8%, respectively. These values were significantly higher than the AgNO_3_ and Ce6 control groups. The authors argued that the presence of bacteria-specific ligands allowed close interaction of the nanocomposites and bacteria surface, followed by the triggered Ag^+^ release that disrupted the membrane proteins, compromising the bacterial cell membrane permeability.

Hou and coworkers employed polymeric micelles (PM) with PpIX located in the core and AgNPs decorating the polymeric shell (PM@PpIX@AgNP) [20]. The design consisted of self-assembled amphiphilic diblock copolymer poly(aspartic acid)-block-poly(ε-caprolactone) (PAsp-b-PCL) micellar nanoparticles with a hydrophobic core containing PpIX. The Ag^+^ ions were introduced within the shell of these polymeric micelles which were eventually reduced to generate in situ AgNPs. The zeta potential was reduced from −55 mV (PM only) to −17 mV (PM@PpIX@AgNPs) with a hydrodynamic size of 100–110 nm. PM@PpIX and PM@PpIX@AgNP showed similar ROS generation ability evaluated by using a singlet oxygen generation probe. The antibacterial activity assessed in bioluminescent *S. aureus* XEN36 (MDR strain) indicated that PM@PpIX@AgNP showed a better antibacterial response than PM@PpIX at concentrations lower than 200 µg/mL. These results were further validated using a LIVE/DEAD confocal study, which showed that the PM@PpIX@AgNP after irradiation completely eliminated the bacteria. The in vivo antibacterial activity was further tested using the bioluminescent *S. aureus* Xen36-infected mouse model. PM@PpIX@AgNP + light exhibited the strongest eradication effect of *S. aureus* XEN36 with no area of infection visible post-5 days.

### 4.4. Sequential Approach of PSs and AgNPs

Studies have also explored introducing PS or AgNPs to interact with bacteria sequentially. For example, PSs is first used then followed by AgNPs to demonstrate the synergy between both alternatives [65,69]. Nakonieczna and group reported that the initial inactivation of *S. aureus* via the PDI (PpIX) effect made them susceptive to AgNP antimicrobial action, resulting in up to a 7 log reduction [69]. This sequential PDI and AgNP combination was also demonstrated in methicillin-resistant *Staphylococcus aureus* 4591 (*MRSA*) and extended-spectrum beta-lactamase-producing *Klebsiella pneumoniae* 2486 (ESBL-KP) [65].

In summary, the PS-AgNP conjugate synthesis can be accomplished via physical mixing and covalent conjugation. The chemical conjugation offers an advantage in terms of stable conjugates and maintaining distance between the AgNPs and PSs. The antibacterial synergy of PS-AgNP was associated with the increased generation of ROS and release of Ag^+^. Resistant and non-resistant forms of *S. aureus* and *E. coli* were found to be the most studied bacterial species. Finally, the incorporation of PS-AgNP conjugates within a micro or nanoscale platform such as fabric, polymeric nanomaterials, or doped nanomaterials allowed for the generation of light-activated nanocomposites for potential multi-modal antibacterial applications.

## 5. Combination of Photosensitizers and Silver Nanoparticles for Inhibition or Elimination of Biofilms

A biofilm is a slime layer composed of the surface-attached aggregates of microorganisms [70]. Biofilms are more complex environments composed of extracellular polysaccharide substances (EPS), proteins, extracellular DNA, and bacterial cells [71]. AgNPs have been used as a preventative measure in implants and prosthetics to inhibit biofilm formation [11,72,73,74,75]. Reports have suggested that AgNPs tested in vitro in planktonic *K. pneumoniae*, *S. aureus*, and *E. coli* disrupt cell-cell adhesion and inhibit biofilm formation [76,77]. Incorporating PS into the AgNPs can further assist in EPS disruption, making the PS-AgNP conjugates excellent candidates for biofilm inhibition or elimination. Table 2 summarizes the PS-AgNP conjugates reported in the literature for the inhibition or elimination of biofilms.

Misba et al. were the first ones to test the biofilm inhibition activity of toluidine blue (TBO)-AgNP conjugates in *Streptococcus mutans* using a 630 nm light source (130 mW/cm^2^) [78]. Two types of TBO-AgNP conjugates were synthesized by the electrostatic interaction of TBO with dextran-capped AgNP (AgNP_dex_) or citrate-capped AgNP (AgNP_cit_). These nanoparticles showed a 46% and 62% reduction in biofilm presence at a concentration of 10 µg/mL, respectively. The authors also reported that both types of TBO-AgNP conjugates inhibited biofilm formation by 99% whereas TBO by itself reduced it by only 70%. Cellular ROS assessment revealed that the TBO-AgNP_dex_ and TBO-AgNP_cit_ mainly generated hydroxyl radicals (OH•) and resulted in a 12-fold and 10-fold increase, respectively, in ROS generation compared with AgNPs. Aydin and co-workers tested a mixture of TBO-AgNP conjugates in 120 teeth samples layered with 3-week-old biofilms. After treatment, a 2.71 average log reduction in bacteria was observed post-60 s irradiation [79]. However, it should be noted that the control treatment with 2.5% NaOCl showed a higher bacterial reduction of 4.29 log. NaOCl is suggested as the primary irrigation solution in endodontics due to its wide-spectrum antimicrobial efficiency as well as its capability in dissolving organic substances [80].

**Table 2 molecules-30-00626-t002:** List of PS-AgNP conjugates tested for inhibition or elimination of biofilms.

Photosensitizer	NP Design	Biofilm Strain	Type of Biofilm Tested (Inhibition vs. Elimination)	Reference
Toluidine blue (TBO)	Electrostatic interaction with AgNPs	*Enterococcus* *faecalis* *Streptococcus mutans*	Inhibition (tested in planktonic bacteria)	[78,79]
Methylene blue	Electrostatic interaction with AgNPs	*S. aureus* *P. aeruginosa*	Inhibition (tested in planktonic bacteria)	[81]
Curcumin	Physical mixture	*Pseudomonas aeruginosa*	Inhibition (tested in planktonic bacteria)	[82]
Chlorin e6	PEI-conjugated Ce6 electrostatically linked with AgNP	*E. coli* *S. aureus*	Inhibition (tested in planktonic bacteria) and Elimination (tested in mature biofilm)	[83]

Methylene blue (MB) is also a phenothiazinium-based cationic dye similar to TBO, which has been widely used to eliminate bacteria. Parasuraman et al. reported a study where MB was physically loaded via electrostatic interactions on the surface of AgNPs. The antibacterial and antibiofilm activity of MB-AgNPs was tested in ESKAPE pathogens such as *S. aureus and P. aeruginosa* biofilms [81]. MB-AgNPs showed a UV-Vis peak at 668 nm associated with MB, confirming its successful conjugation. The bacterial uptake studies showed that MB-AgNPs had greater localization within the bacteria than MB. In *P. aeruginosa* and *S. aureus*, 75.4% and 78.33% of MB was internalized using the MB-AgNPs, whereas only 43.6% and 35.5% of MB was internalized without conjugation. The minimum inhibitory concentration (MIC) value in *S. aureus* and *P. aeruginosa* was estimated to be 125 µg/mL. This MIC had two-fold and four-fold higher antimicrobial activity than AgNPs and free MB, respectively. At the same MIC concentration, 61.41% and 69.15% biofilm inhibition was observed in light-irradiated MB-AgNPs in *P. aeruginosa* and *S. aureus*, respectively. The authors reported that MB-AgNPs showed higher levels of ROS generated by MB-AgNPs, confirming that oxidative stress drove their antibacterial and antibiofilm action.

Curcumin is another PS that has been combined with AgNPs. Ghasemi et al. tested a mixture of curcumin and AgNPs in *P. aeruginosa* planktonic bacteria and biofilms [82]. The mixture containing 200 and 20 µg/mL of curcumin and AgNPs, respectively, was irradiated for 10 min with blue light in the presence of *P. aeruginosa*. An inhibition of 85% of biofilm formation was determined, whereas curcumin and AgNPs only showed 50% and 60% biofilm inhibition, respectively. The mixture of curcumin and AgNPs showed the highest generation of ROS, which contributed to the superior biofilm inhibition capability. The authors reported that the effect induced by this mixture was more effective in inhibiting biofilm formation than removing adhered biofilm.

The above studies have mainly focused on demonstrating the use of PS-AgNP systems for inhibiting biofilm formation. However, Sun and collaborators developed a nanocomposite containing chlorin e6 (Ce6)-modified PEI associated with AgNPs to demonstrate the synergistic effect of eliminating biofilms from Gram-positive (*S. aureus*) and Gram-negative (*E. coli*) bacteria [83]. The authors showed that the presence of AgNPs raised the production of singlet oxygen and ROS due to the SPR effect. The nanocomposite, with a size of 60 nm in diameter, was tested in planktonic *S. aureus* and *E. coli* under 660 nm irradiation, using 20 mW/cm^2^ for 20 min at a 1 mg/mL concentration. The antibacterial action was more pronounced in *E. coli* with 100% elimination. The authors argued that this was due to the higher sensitivity of Gram-negative bacteria toward Ag^+^ ions. In the case of *S. aureus*, 99.9% elimination was achieved, corresponding to a ~2 log reduction. The penetration of the variable mass ratio of the PEI-Ce6/AgNP, i.e., 0.3 (negative charge; smaller diameter), 1.5 (a positive charge; larger diameter, 240 nm), and 3.0 (a positive charge; smaller diameter, 70 nm) within the *S. aureus* biofilm was evaluated. This approach showed that the highest ratio, i.e., 3.0 with a positive charge (+35 mV) and smaller hydrodynamic size (70 nm), showed better infiltration within the EPS matrix than other ratios, indicating that both size and surface charge plays a crucial role. The PEI-Ce6/AgNPs showed increased biofilm elimination in the preformed biofilms, resulting in a 2–3 log reduction in bacterial growth in both *S. aureus* and *E. coli*. Nevertheless, in the case of *E. coli*, the authors had to employ a membrane-damaging antibiotic polymyxin B (PMB) in addition to the nanocomposite to show improved results. As a proof of concept for wound healing applications, these nanocomposites were tested in vivo in mice models with *S. aureus* skin infection. The nanocomposite and light-treated mice showed faster wound healing, with the complete eradication of biofilm achieved on the 12th day.

In summary, fewer PS-AgNPs conjugates have been tested in biofilms than bacteria. Most of these PS-AgNP combinations, comprising photosensitizers such as toluidine blue, methylene blue, and curcumin reportedly displayed the inhibition of biofilm formation (Figure 3). There has been only one study which has studied the penetration of PS-AgNP within biofilms as well as the red-light-induced eradication of biofilms (*S. aureus* and *E. coli*) [83]. This study also pointed to the fact that smaller and positively charged PS-AgNPs show greater penetration within biofilms. The PDI/AgNP synergistic effect has been demonstrated in multiple biofilm-forming bacterial strains such as *Pseudomonas aeruginosa*, *Enterococcus faecalis*, and *Streptococcus mutans* [78,79,81].

## 6. Conclusions

AgNPs are highly relevant in the fight against bacteria and biofilms due to their potent properties. AgNPs show broad-spectrum antimicrobial activity with unique bacterial cell death mechanisms, including ROS generation, membrane disruption, protein dysfunction, and DNA damage. Additionally, AgNPs effectively prevent biofilm formation and disrupt existing biofilms by altering bacterial adhesion and damaging the biofilm matrix. When combined with PSs, AgNPs can generate additive or synergistic effects, making them more potent against bacteria, resistant bacterial strains, and biofilms.

Light-activated PS-AgNPs offer several advantages for eliminating bacteria and other pathogens, such as enhanced antimicrobial activity associated with the generation of ROS. Moreover, the bactericidal effect can be controlled by adjusting the light source and exposure, which allows for targeted activation, reducing the risks of unwanted side effects on non-target cells or tissue. PS-AgNPs can also help in reducing the likelihood of bacteria developing resistance because the combination of light, PS, and AgNPs makes it harder for bacteria to adapt. Several chemical approaches have been explored to synthesize PS-AgNPs, like physical association, chemical conjugation, and composites. PS-AgNPs, fabricated by the physical interaction of PSs and AgNPs, are relatively easy to prepare but lack stability and reproducibility. PS-AgNP composites are complex systems that confer unique benefits such as stability, target delivery, and potential multi-modal antibacterial applications. However, their complexity and reproducibility are major challenges. The chemically conjugated PS-AgNP approach offers several advantages, including stability, reproducibility, and target delivery.

There are only a few reports on the use of light-activated PS-AgNPs to inhibit or eliminate biofilms. These reports have been focused on biofilm prevention with promising results to inhibit the formation of biofilms. Nevertheless, only one paper was found that showed the use of PS-AgNPs to combat biofilms. We anticipate that several advantages associated with light-activated PS-AgNPs can improve the elimination of biofilms, for wound healing applications. The generation of ROS by PS-AgNPs can break down the biofilm matrix more effectively, resulting in the killing of embedded bacteria. Due to the tunable physicochemical properties of PS-AgNPs, these can penetrate biofilms deeper, which leads to a more thorough eradication. Moreover, the light activation mechanism can enhance the permeability of biofilms, allowing other drugs like antibiotics to penetrate more effectively. Finally, activating PS-AgNPs with light allows for the precise targeting of biofilms, minimizing damage to surrounding healthy tissue and reducing potential side effects.

## 7. Future Outlook

Light-activated PS-AgNPs are envisioned for use in various settings, including medical devices, coatings, hospitals surfaces, and textiles. These light-activated PS-AgNPs can potentially be considered for the treatment of chronic wounds and wound healing applications. Their ability to be activated on demand makes them suitable for dynamic environments where constant antimicrobial action is needed. However, their use should be limited to topical or external applications to avoid long term toxicity associated with AgNPs/Ag^+^. Overall, PS-AgNPs offer a promising solution to combat infections and biofilms, especially in an era when antibiotic resistance is a growing concern.

## Figures and Tables

**Figure 1 molecules-30-00626-f001:**
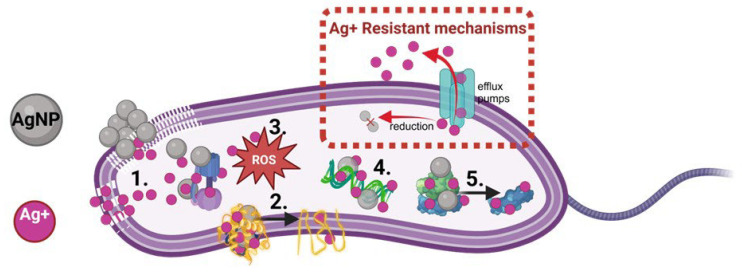
The antibacterial mechanisms of AgNPs and Ag^+^ ions. (1) Binding and disruption of the cell membrane. (2) Denaturation of the membrane-bound and intracellular proteins. (3) Interfering with intracellular metabolic pathway function and reactive oxygen species (ROS) generation. (4) Interfering with the DNA replication process. (5) Disassembly of the ribosomal subunit. (Dashed frame) Ag^+^ resistance in bacteria is associated with reducing Ag^+^ to less toxic forms and eliminating intracellular Ag^+^ via efflux pumps.

**Figure 2 molecules-30-00626-f002:**
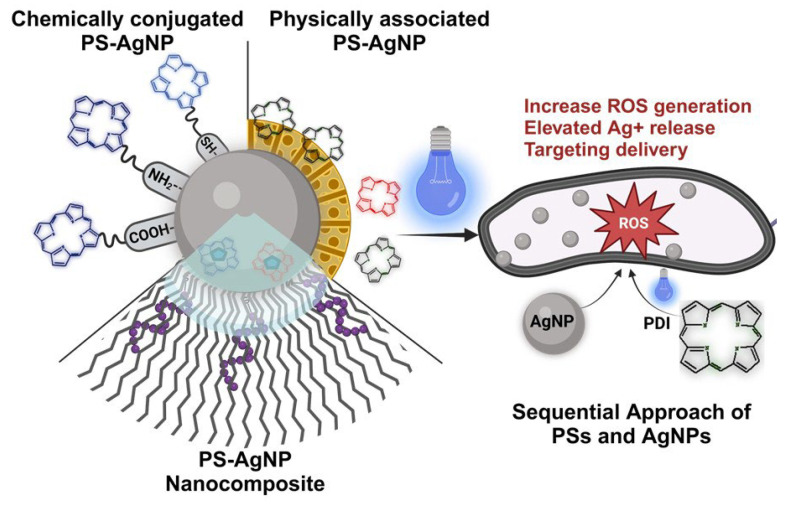
Summary of the different approaches used to combine PSs and AgNPs, including chemically conjugated PS-AgNPs, physically associated PS-AgNPs, PS-AgNP composites, and sequential treatment with PS and AgNPs. All these systems have been tested in non-resistant and resistant bacterial strains.

**Figure 3 molecules-30-00626-f003:**
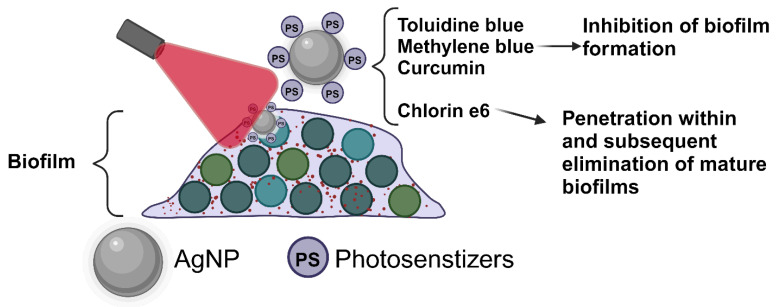
The summary of PS-AgNP nanocomposite designs for the light-activated inhibition and elimination of bacterial biofilms.

**Table 1 molecules-30-00626-t001:** List of PS-AgNPs nanocomposite designs reported for eliminating resistant and non-resistant bacterial strains.

Nanoparticle Design	Photosensitizer	Conjugation Chemistry	Bacterial Species Tested	Light Source	Reference
AgNP core with mesoporous silica shell (40–50 nm/2–17 nm shell thickness)	Hematoporphyrin IX (HPIX) meso-porphyrin IX (PIX) meso-tetra(4-carboxyphenyl) porphine (TCPP) Cu(II) mesotetra(4-carboxyphenyl) porphine (Cu-TCPP) tris(2,20-bipyridyl) dichlororuthenium(II) hexahydrate (RuBPy) Rose Bengal (RB)	Adsorption	Multidrug-resistant *Staphylococcus aureus* (ATCC BAA-44) *Staphylococcus epidermidis* (ATCC 35984) *Escherichia coli* (ATCC 35218) *Acetobacter baumannii* (drug resistant) (ATCC 19606) *Trichophyton rubrum* (ATCC 28188)	White light; 408 mW/cm^2^; 300 mW/cm^2^	[54,60,61]
Silver nanoparticles	Methylene blue	Physical mixture	*Streptococci* *Serratia marcescens* (SM), *Escherichia coli*, *Klebsiella pneumoniae* (KP), *Pseudomonas aeruginosa* (PA), *Enterobacter cloacae* (EC)	Diode Red laser (660 nm; 180 J/cm^2^) LED Red (660 nm; 6.8 mW; 180 J/cm^2^)	[62,63]
Silver nanoparticles	Zn-meso-5,10,15,20-tetra(4-pyridyl) Zn-meso-5,10,15,20-tetrathienyl Zn-meso-5-(4- hydroxyphenyl)- 10,15,20-tris(2-thienyl)	Electrostatic self-assembly	*Staphylococcus aureus*	595 nm LED (15 min, 40 J/cm^2^)	[64]
Silver nanoparticles	TMPyP (5,10,15,20– tetrakis (N-methylpyriminium-4-yl) porphyrin)	Physical mixture	Methicillin-resistant *Staphylococcus aureus* (MRSA), extended-spectrum beta-lactamases-producing (ESBL) *Klebsiella pneumoniae*	LED (414 nm; 54 mW/cm^2^	[65]
AgNP (spherical, triangular, and cubic)	Phthalocyanines	Coordination via NH_2_ and COOH groups of cysteinylmoiety	*S. aureus* (ATCC 6538)	Visible light (300 W, 90 min)	[48]
AgNP	Protoporphyrin IX	NHS-EDC chemistry for linking cysteamine	*MRSA* (ATCC BAA 44 strain), Wild type *MDR E. coli*	White light (400–700 nm; 56 ± 2 mW/cm^2^)	[21]
AgNP in situ synthesized with cellulose fabric	Zinc pthalocyanines	PS linked to cellulose fabric via EDC chemistry. No direct link with AgNPs	*E. coli* (ATCC 8739), *S. aureus* (ATCC 6538), *MRSA* (ATCC 33591)	LED light (75 mW/cm^2^, 10 min, 660 nm)	[59]
Silver nanoparticles (2 nm) (embedded in PMMA fibers)	Tetraphenylporphyrin (TPP)	Electrospinning; No direct link with AgNPs	*Staphylococcus epidermidis*, *Enterococcus faecalis*	LED (405 nm; 3 h; 400 mW/cm^2^	[66]
Polymeric nanoparticles with PpIX core and AgNP embedded within polymeric shell	Protoporphyrin (located in core)	Self-assembly of PpIX with co-block polymer. No direct link with AgNPs	MDR *S. aureus* Xen36	Laser, 635 nm, 0.25 W/cm^2^, 10 min	[20]
Silver nanoparticles + Copper-doped MSN	Curcumin	Adsorption	*Escherichia coli*	Blue LED light 470 nm (72 J/cm^2^)	[67]
Silver nanoparticle with polydopamine (PDA) shell	Chlorin e6 (Ce6)	Ce6 functionalized on AgNP via PDA shell	*MRSA (ATCC 29213)*, *E. coli (ATCC 25922)*	Red laser (655 nm, 300 mW/cm^2^, 10 min)	[58]

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
