# Peer review of "Light-Activable Silver Nanoparticles for Combatting Antibiotic-Resistant Bacteria and Biofilms"

_molecules, 2025, doi:10.3390/molecules30030626_

Round 1

Reviewer 1 Report

Comments and Suggestions for Authors

This review summarizes the PS and Ag NPs synthetic strageties reported in the literature for the inactivation of resistant and non-resistant bacterial strains and the elimination or inhibition of biofilms. This review provide a new vision of the potential possibilities of the nanomaterials. However, there are some questions that need to clarify:

1. Line 80, what's the PDI?

2. Line 120. The physicochemical properties of Ag NPs  play a crucial role in their antibacterial. Does  the synthetic methods has the influence  of the antibacterial?

3. Part 3. Please further demonstrate the influence of the SPR of Ag NPs on their antibacterial.

4. Part 4.  Table 1 could be very useful for the reader as a nice summary for this review. In the main text, the description should be presented in conjunction with the Table.1. It is recommended to adopt a more logical approach of this part writing. 

5.  Part 5. "The authors showed that the presence of Ag NPs raised the production of singlet oxygen and ROS due to the SPR effect." Please further explain it.

6. Table 2. The title of table.2 should be "List of PS-Ag NPs conjugates texted for inhibition or elimination of biofilms". The PS-Ag NP nanocomposites is not accurate.

Author Response

Reviewer #1:

“This review summarizes the PS and Ag NPs synthetic strageties reported in the literature for the inactivation of resistant and non-resistant bacterial strains and the elimination or inhibition of biofilms. This review provide a new vision of the potential possibilities of the nanomaterials. However, there are some questions that need to clarify:”

We would like to thank Reviewer #1 for his/her helpful comments to improve our work.

“1. Line 80, what's the PDI?”

Response: The term PDI was defined in line 72-73.

“2. Line 120. The physicochemical properties of Ag NPs  play a crucial role in their antibacterial. Does  the synthetic methods has the influence  of the antibacterial?”

Response: Yes, it does. The synthetic method define the physicochemical properties of the AgNPs, which have a direct impact on their antibacterial capability. Properties such as surface charge, capping agent, particle size, and shape play a key role. Our review discusses these factors and their subsequent effect on antibacterial properties in sections 2 and 4.  

“3. Part 3. Please further demonstrate the influence of the SPR of Ag NPs on their antibacterial.”

Response: Section 3 of the review provided one example of the use of SPR associated with AgNPs to eliminate bacteria, Reference #40. We have added another paper to further illustrate this phenomenon in line 167: “Vasilkov et. al. reported that laser excitation of AgNP composites (bandages) at the SPR wavelength led to 15-24 % CFU reduction in bacterial species [43].” 

“4. Part 4.  Table 1 could be very useful for the reader as a nice summary for this review. In the main text, the description should be presented in conjunction with the Table.1. It is recommended to adopt a more logical approach of this part writing.”

Response: Thanks for your suggestion, we agree that following the order depicted in Table 1 would create a coherent narrative for Section 4. However, we thought that by using the method used to incorporate or associate the photosensitizers into the AgNPs as a sub-section offers a more instructive approach for the reader.

“5.  Part 5. "The authors showed that the presence of Ag NPs raised the production of singlet oxygen and ROS due to the SPR effect." Please further explain it.”

Response: Reference #44 was included in line 169 to further explain this concept:” This can be attributed to SPR-associated electronic changes in AgNP or localized heating that enhances their antibacterial effect. Like most noble metals, the SPR in AgNP induces elevated absorption of the incident light, which is associated with ROS generation [44].”

“6. Table 2. The title of table.2 should be "List of PS-Ag NPs conjugates texted for inhibition or elimination of biofilms". The PS-Ag NP nanocomposites is not accurate.”

Response: The title was changed based on reviewer’s suggestion.

Thank you for your consideration. I look forward to your response.

Reviewer 2 Report

Comments and Suggestions for Authors

Summary: The manuscript reviews the prior literature on light activable AgNPs to combat antibiotic resistance and biolfilms. The literature reports show the augmentation of AgNP with PSs, causing enhanced antimicrobial action through light activated mechanism. The manuscript also discusses the various approaches for synthesis of PS conjugated AgNP.

The mini-review topic is likely to attract the attention of the Molecules audience. The scope of work covered is appropriate. However, the manuscript should be revised to reach an acceptable level of accuracy, precision, and clarity before it can be accepted for publication. One recurring issue is that the manuscript uses qualitative terms and descriptions in places where quantitative terms and figures would be more appropriate and informative. Furthermore, the manuscript needs a thorough proof-reading to make sure the formatting is uniform and acceptable by journal standards.

A partial list follows below, but it is suggested that the authors take a critical approach to revising and proofreading their manuscript with this feedback in mind.

Recommendation: Reconsider after major revisions.

·       Page 1, Line 35: Please change “targets, (d)” to “targets, and (d)”.

·       Page 2, Line 50: “silver nanoparticles (AgNPs) have attracted 49 more interest due to their small size”. Please mention the size range of the AgNPs for better clarity to the reader.

·       Page 3, Line 105: Please change the font for “highly associated” to make it uniform.

·       Page 5, Line 190: “The combination of PSs and AgNPs reported in the literature can be divided into four main categories (Figure 2)”. The Figure 2 only depicts the three categories mentioned as (1), (2), and (3). Please include the category (d) into Figure 2 as well.

·       Page 5, Line 193: Please make the numbering of various categories of PS and AgNP combination uniform. Change “d) sequential combinations of PSs and AgNPs” to “ 4) sequential combinations of PSs and AgNPs”.

·       Page 5, Table 1: Please delete the un-needed citation present in the “Nanoparticle Design” column.

·       Page 6, Table 1: Please change “NH2” to “NH2”.

·       Page 7, Line 204: Please provide appropriate numbering for the respective heading and sub-headings. The title “PSs physically associated with AgNPs” is missing the heading number.

·       Page 7, Line 214: Is “MB” used as an acronym for methylene blue? If yes, please mention it in the discussion for better clarity to the reader.

·       Page 7, Line 223: “AgNPs/TPP/PMMA nanofibers exhibited a significantly higher reduction of bacteria than TPP/PMMA”. How much greater reduction is observed for AgNPs/TP/PMMA? Is the difference observed statistically significant?

·       Page 7, Line 244: Please update the heading number for “1. PS chemically conjugated to AgNPs”.

·       Page 7, Line 247: “A chemical bond between PS and AgNP also allows control of distance and target delivery”. How does the distance between the PS and AgNP surface affect the biological activity? Does the different length of linkers have an effect of the biological activity of PS containing AgNPs.

·       A discussion on the above-mentioned topics is suggested to provide better clarity to reader.

·       Page 7, Line 279: Is “PpIX” used as an acronym for porphyrin. If yes, please mention in discussion for better clarity to the reader.

·       Page 7, Line 280: “Due to its ionic composition, this effect was more pronounced (25-fold) in Dulbecco’s phosphate buffer solution (DPBS) than in nanopure water”. Can the author add a brief discussion of how does the ionic strength of the medium contribute to greater release of Ag from AgNP conjugates?

·       Page 9, Line 310: “silver nanoparticles (SNP)”. AgNP has been used as the acronym for silver nanoparticles in the manuscript. Please do necessary changes after thorough proofreading to maintain uniformity in acronyms used.

·       Page 9, Line 338: “hydrophilic self-assembled amphiphilic diblock copolymer poly(aspartic acid)-block-poly(ε-caprolactone) (PAsp-b-PCL)”. The polymer nomenclature used as “hydrophilic self-assembled amphiphilic diblock copolymer” is not clear. Poly(aspartic acid) is a hydrophilic polymer and poly(caprolactone) is a hydrophobic polymer, thus the diblock copolymer can be considered to have an amphiphilic nature. Please rewrite the polymer nomenclature for better clarity.

·       Page 9, Line 352: “1. Sequential approach of PSs and AgNPs”. All the headings in manuscript are numbered as “1.”

·       Please follow a numerical sequence for all the headings/sub-heading in the manuscript for uniformity.

·       The manuscript discusses the synergistic effect of AgNP containing PSs towards bactericidal action. The approach provides benefits in the form of lower antimicrobial resistance and higher bactericidal effect, as described in the manuscript.

Can the photodynamic therapy approach be used to achieve spatiotemporal control over the target site? Also, what are the effects of different light sources (e.g., near IR, UV, visible) over the efficacy of the photodynamic therapy by AgNP containing PS?

It is suggested to add a discussion on the above topics to make the manuscript more insightful for interested readers.

·       The manuscript summarizes the synthesis strategies for PS-AgNP synthesis. Can the authors add an additional figure to present the synthetic strategies in the form of chemical reaction schemes with chemical structures and reaction conditions? Presenting the synthetic strategies in the form of reactions schemes can make the discussion more effective and approachable to interested readers.

·       References: A through proofreading is recommend, keeping in mind the comments below.

Please delete the “issue number” from the references. The manuscript includes some references with issue number and others without issue number. Please maintain uniformity in format.

Delete “Volume” before the volume number. E.g. Reference 5. Journal of Nanomedicine 2017, Volume 12, 1227-1249.

Use acronym for the journal name instead of using the full name. E.g. “Nanomed. J.” is the accepted acronym for “Journal of Nanomedicine”.

Add the missing page number for the references. E.g. Reference 17. International Journal of Molecular Sciences 2019, 20 (1). (page number missing)

Delete “pp” and ‘vol” from the page and volume numbers. E.g. Reference 18. Eds. Academic Press: 2022; Vol. 79, pp 65-108.

Author Response

Reviewer #2:

“The manuscript reviews the prior literature on light activable AgNPs to combat antibiotic resistance and biolfilms. The literature reports show the augmentation of AgNP with PSs, causing enhanced antimicrobial action through light activated mechanism. The manuscript also discusses the various approaches for synthesis of PS conjugated AgNP.

The mini-review topic is likely to attract the attention of the Molecules audience. The scope of work covered is appropriate. However, the manuscript should be revised to reach an acceptable level of accuracy, precision, and clarity before it can be accepted for publication. One recurring issue is that the manuscript uses qualitative terms and descriptions in places where quantitative terms and figures would be more appropriate and informative. Furthermore, the manuscript needs a thorough proof-reading to make sure the formatting is uniform and acceptable by journal standards.

A partial list follows below, but it is suggested that the authors take a critical approach to revising and proofreading their manuscript with this feedback in mind.”

We would like to thank Reviewer #2 for his/her helpful comments to improve our work.

“1. Page 1, Line 35: Please change “targets, (d)” to “targets, and (d)”.”

Response: Changed.

“2. Page 2, Line 50: “silver nanoparticles (AgNPs) have attracted 49 more interest due to their small size”. Please mention the size range of the AgNPs for better clarity to the reader.”

Response: Size added: “(<100 nm)”

“3. Page 3, Line 105: Please change the font for “highly associated” to make it uniform.”

Response: Changed.

“4. Page 5, Line 190: “The combination of PSs and AgNPs reported in the literature can be divided into four main categories (Figure 2)”. The Figure 2 only depicts the three categories mentioned as (1), (2), and (3). Please include the category (d) into Figure 2 as well.”

Response: Thanks for highlighting this point. Figure 2 has been modified to address this point.

“5. Page 5, Line 193: Please make the numbering of various categories of PS and AgNP combination uniform. Change “d) sequential combinations of PSs and AgNPs” to “ 4) sequential combinations of PSs and AgNPs”.”

Response: This sentence have been deleted by addressing reviewer’s previous comment.

“6. Page 5, Table 1: Please delete the un-needed citation present in the “Nanoparticle Design” column.”

Response: There are not citations on this column. The numbers (“(40-50 nm/ 2-17 nm shell”) refer to nanoparticle specifications.

“7.   Page 6, Table 1: Please change “NH2” to “NH2””

Response: Changed.

“8. Page 7, Line 204: Please provide appropriate numbering for the respective heading and sub-headings. The title “PSs physically associated with AgNPs” is missing the heading number.”

Response: The mentioned title is localized in line 209 “4.1. PSs Physically Associated with AgNPs”

“9. Page 7, Line 214: Is “MB” used as an acronym for methylene blue? If yes, please mention it in the discussion for better clarity to the reader.”

Response: Changed.

“10. Page 7, Line 223: “AgNPs/TPP/PMMA nanofibers exhibited a significantly higher reduction of bacteria than TPP/PMMA”. How much greater reduction is observed for AgNPs/TP/PMMA? Is the difference observed statistically significant?”

Response: The following sentence was added in line 229: “The disk-diffusion method showed < 100mm2 for TPP/PMMA as compared to > 400 mm2 for AgNPs/TPP/PMMA.” No statistical analysis was provided in the original paper.

“11. Page 7, Line 244: Please update the heading number for “1. PS chemically conjugated to AgNPs”.”

Response: Heading number (4.2) is correct. As indicated in a previous response, heading number 4.2 is in line 209.

“12. Page 7, Line 247: “A chemical bond between PS and AgNP also allows control of distance and target delivery”. How does the distance between the PS and AgNP surface affect the biological activity? Does the different length of linkers have an effect of the biological activity of PS containing AgNPs.”

Response: Those are very good point brought by the reviewer; however, we could not find any specific study related to PS and AgNPs reported on this topic.

“13. A discussion on the above-mentioned topics is suggested to provide better clarity to reader.”

Response: Since there are not specific papers published on those subjects, we cannot write a discussion on them.

“14. Page 7, Line 279: Is “PpIX” used as an acronym for porphyrin. If yes, please mention in discussion for better clarity to the reader.”

Response: Added in line 272: “protoporphyrin IX (PpIX)”

“15. Page 7, Line 280: “Due to its ionic composition, this effect was more pronounced (25-fold) in Dulbecco’s phosphate buffer solution (DPBS) than in nanopure water”. Can the author add a brief discussion of how does the ionic strength of the medium contribute to greater release of Ag from AgNP conjugates?”

Response: The following sentence has been added in line 288: “The authors hypothesized that the presence of Cl- ions can account for some of the Ag+ release differences between water and DPBS.”

“16. Page 9, Line 310: “silver nanoparticles (SNP)”. AgNP has been used as the acronym for silver nanoparticles in the manuscript. Please do necessary changes after thorough proofreading to maintain uniformity in acronyms used.”

Response: Changed.

“17. Page 9, Line 338: “hydrophilic self-assembled amphiphilic diblock copolymer poly(aspartic acid)-block-poly(ε-caprolactone) (PAsp-b-PCL)”. The polymer nomenclature used as “hydrophilic self-assembled amphiphilic diblock copolymer” is not clear. Poly(aspartic acid) is a hydrophilic polymer and poly(caprolactone) is a hydrophobic polymer, thus the diblock copolymer can be considered to have an amphiphilic nature. Please rewrite the polymer nomenclature for better clarity.”

Response: The sentence has been rewritten as follows in line 343: “The design consisted of self-assembled amphiphilic diblock copolymer poly(aspartic acid)-block-poly(ε-caprolactone) (PAsp-b-PCL) micellar nanoparticles with a hydro-phobic core containing PpIX.” The nomenclature has been kept as originally reported in the paper to have consistency.

“18. Page 9, Line 352: “1. Sequential approach of PSs and AgNPs”. All the headings in manuscript are numbered as “1.””

Response: Sorry for this issue, but in our paper version all the headings are correctly numbered. In this case, “Sequential Approach of PSs and AgNPs” is heading 4.4.

“19. Please follow a numerical sequence for all the headings/sub-heading in the manuscript for uniformity.”

Response: As indicated in our previous response, all the headings and sub-headings have been correctly numbered in our version of the manuscript.

“20. The manuscript discusses the synergistic effect of AgNP containing PSs towards bactericidal action. The approach provides benefits in the form of lower antimicrobial resistance and higher bactericidal effect, as described in the manuscript.”

Response: Not clear if any suggestion was written here.

“21. Can the photodynamic therapy approach be used to achieve spatiotemporal control over the target site? Also, what are the effects of different light sources (e.g., near IR, UV, visible) over the efficacy of the photodynamic therapy by AgNP containing PS?”

Response: Thanks for the suggested topics/questions. Some of these concepts have been illustrated in section 4, but we consider that those topics are more relevant for photodynamic therapy (PDT) than photodynamic inactivation (PDI). Therefore, it would be out of the scope of this review to include them.

“22. It is suggested to add a discussion on the above topics to make the manuscript more insightful for interested readers.”

Response: See our response above.

“23. The manuscript summarizes the synthesis strategies for PS-AgNP synthesis. Can the authors add an additional figure to present the synthetic strategies in the form of chemical reaction schemes with chemical structures and reaction conditions? Presenting the synthetic strategies in the form of reactions schemes can make the discussion more effective and approachable to interested readers.”

Response: Thanks for the suggestion. However, we have already listed the type of chemistry used for conjugation in Table 1. Moreover, adding all the experimental conditions and chemical structures for 5-7 reactions in a figure it may be confusing.

“24. References: A through proofreading is recommend, keeping in mind the comments below.

Please delete the “issue number” from the references. The manuscript includes some references with issue number and others without issue number. Please maintain uniformity in format.

Delete “Volume” before the volume number. E.g. Reference 5. Journal of Nanomedicine 2017, Volume 12, 1227-1249.

Use acronym for the journal name instead of using the full name. E.g. “Nanomed. J.” is the accepted acronym for “Journal of Nanomedicine”.

Add the missing page number for the references. E.g. Reference 17. International Journal of Molecular Sciences 2019, 20 (1). (page number missing)

Delete “pp” and ‘vol” from the page and volume numbers. E.g. Reference 18. Eds. Academic Press: 2022; Vol. 79, pp 65-108.”

Response: All the corrections have been carried out.

Thank you for your consideration. I look forward to your response.

Reviewer 3 Report

Comments and Suggestions for Authors

[Major Issues]

Comment 1: Effects on AgNPs’ accumulation in the body

Through the combination of AgNPs and photosensitizers, I understood that light-activated biofilm is a great strategy for antibiotic-resilient bacteria. But I have concerns with AgNPs.

1) Figure 1 explains the antibiotic mechanism well, but is there a problem with AgNPs or Ag+ ions unintentionally staying in the body? According to the author, photosensitizers improve bioavailability by combining with AgNPs, but I am worried that AgNPs separated from photosensitizer will not be toxic in the body.

2) Or is there a strategy to solve the problem of AgNPs or Ag+ ions staying in the body?

3) According to the previous studies (https://pmc.ncbi.nlm.nih.gov/articles/PMC3878143/), long-term exposure to AgNPs resulted in cytotoxicity. Therefore, I recommend that you comment on the problems caused by the long-term use of PS-AgNP biofilms.

4) According to the previous studies (https://doi.org/10.1073/pnas.1911734116), small-sized gold nanoparticles are degraded and recrystallized in unexpected ways within the cell. As you mentioned that the antibacterial effect is greater in small AuNPs under 30 nm, I am worried that there is a problem that degrades faster in the body due to the small size. Although this paper may differ with my expectations by using silver nanoparticles, I would like to know if there is another paper that deals with the problem of the easily degrading of small-sized nanoparticles in the body, and if there is, please explain the problems and a solution strategy using scientific results.

[Minor Issues]

1. 3 of 15, Line 105. Please change the font size. ‘highly associated with’

2. Figure 1: Please adjust the size of the numbers representing the order of the antibacterial mechanism to improve visibility.

3. 4 of 15, Line 153, Please correct the typo. (~4 logs to  ~4 log)

4. 4 of 15, Line 204, 244, 285, 352, The numbering in the paragraph (Combination of photosensitizers with ~’) subheading is inconsistent. Please correct it.

Author Response

Reviewer #3:

“This review summarizes the PS and Ag NPs synthetic strageties reported in the literature for the inactivation of resistant and non-resistant bacterial strains and the elimination or inhibition of biofilms. This review provide a new vision of the potential possibilities of the nanomaterials. However, there are some questions that need to clarify:”

We would like to thank Reviewer #3 for his/her helpful comments to improve our work.

“1. Figure 1 explains the antibiotic mechanism well, but is there a problem with AgNPs or Ag+ ions unintentionally staying in the body? According to the author, photosensitizers improve bioavailability by combining with AgNPs, but I am worried that AgNPs separated from photosensitizer will not be toxic in the body.”

Response: Thanks for bringing this point, this is a valid concern; however, in this review, we have mentioned that the PS-AgNPs combination is better suited for wound healing applications or devices. Line 502: “Light-activated PS-AgNPs are envisioned for use in various settings, including medical devices, coatings, hospitals surfaces and textiles. These light activated PS-AgNPs can potentially be considered for treatment of chronic wounds and wound healing ap-plications.” This minimizes the internalization and potential toxicity of AgNPs in the body.

“2. Or is there a strategy to solve the problem of AgNPs or Ag+ ions staying in the body?”

Response: Please see our response above.

“3. According to the previous studies (https://pmc.ncbi.nlm.nih.gov/articles/PMC3878143/), long-term exposure to AgNPs resulted in cytotoxicity. Therefore, I recommend that you comment on the problems caused by the long-term use of PS-AgNP biofilms.”

Response: The following sentence was added to line 506: “However, their use should be limited to topical or external applications to avoid long term toxicity associated with AgNPs/ Ag+.”

“4. According to the previous studies (https://doi.org/10.1073/pnas.1911734116), small-sized gold nanoparticles are degraded and recrystallized in unexpected ways within the cell. As you mentioned that the antibacterial effect is greater in small AuNPs under 30 nm, I am worried that there is a problem that degrades faster in the body due to the small size. Although this paper may differ with my expectations by using silver nanoparticles, I would like to know if there is another paper that deals with the problem of the easily degrading of small-sized nanoparticles in the body, and if there is, please explain the problems and a solution strategy using scientific results.”

Response: Please see our previous responses. Major issues in terms of toxicity for AgNPs will be associated with systemic administration, which is not the case for the applications discussed in this review.

“5. [Minor Issues]

  1. 3 of 15, Line 105. Please change the font size. ‘highly associated with’
  2. Figure 1: Please adjust the size of the numbers representing the order of the antibacterial mechanism to improve visibility.
  3. 4 of 15, Line 153, Please correct the typo. (~4 logs to ~4 log)
  4. 4 of 15, Line 204, 244, 285, 352, The numbering in the paragraph (Combination of photosensitizers with ~’) subheading is inconsistent. Please correct it.”

Response: All minor issues have been addressed.

Thank you for your consideration. I look forward to your response.

Round 2

Reviewer 1 Report

Comments and Suggestions for Authors

I think the manuscript can be accepted in present form.

Reviewer 2 Report

Comments and Suggestions for Authors

Thank you for addressing the previous comments.

Comments on the revised manuscript:

- All the references in the manuscript has been added twice mistakenly. Please rectify this formatting error.

Reviewer 3 Report

Comments and Suggestions for Authors

There are no additional comments for this paper